

# MCNMF-Unet: a mixture Conv-MLP network with multi-scale features fusion Unet for medical image segmentation

Lei Yuan, Jianhua Song and Yazhuo Fan

Key Laboratory of Light Field Manipulation and System Integration Applications in Fujian Province, School of Physics and Information Engineering, Minnan Normal University, Zhangzhou, Fujian, China

## ABSTRACT

Recently, the medical image segmentation scheme combining Vision Transformer (ViT) and multilayer perceptron (MLP) has been widely used. However, one of its disadvantages is that the feature fusion ability of different levels is weak and lacks flexible localization information. To reduce the semantic gap between the encoding and decoding stages, we propose a mixture conv-MLP network with multi-scale features fusion Unet (MCNMF-Unet) for medical image segmentation. MCNMF-Unet is a U-shaped network based on convolution and MLP, which not only inherits the advantages of convolutional in extracting underlying features and visual structures, but also utilizes MLP to fuse local and global information of each layer of the network. MCNMF-Unet performs multi-layer fusion and multi-scale feature map skip connections in each network stage so that all the feature information can be fully utilized and the gradient disappearance problem can be alleviated. Additionally, MCNMF-Unet incorporates a multi-axis and multi-windows MLP module. This module is fully end-to-end and eliminates the need to consider the negative impact of image cropping. It not only fuses information from multiple dimensions and receptive fields but also reduces the number of parameters and computational complexity. We evaluated the proposed model on BUSI, ISIC2018 and CVC-ClinicDB datasets. The experimental results show that the performance of our proposed model is superior to most existing networks, with an IoU of 84.04% and a F1-score of 91.18%.

## INTRODUCTION

In recent years, high-performance methods based on convolutional neural network (CNN) have demonstrated superior performance on many tasks (*Kadry et al., 2022*; *Sun et al., 2022*; *Zamir et al., 2021*; *Li et al., 2021b*; *Ding et al., 2022b*; *Kalake et al., 2022*). Benefiting from the development of CNN, computer vision techniques have been widely used in the field of medical image processing. Image semantic segmentation is an important component of medical image processing, especially accurate and robust medical image segmentation techniques can play a cornerstone role in computer-aided diagnosis and image-guided clinical surgery (*Hatamizadeh et al., 2022*; *Valanarasu & Patel, 2022*; *Xie et al., 2022*).

Corresponding author
Jianhua Song, songjian-hua@mnnu.edu.cn

Image semantic segmentation can be formulated as a typical dense prediction problem, which aims at the pixel-level classification of feature maps. Existing CNN-based medical image segmentation methods mainly rely on fully convolutional neural network (FCNN) (*Isensee et al., 2021*; *Jin et al., 2020*). The most typical of these is the Unet (*Ronneberger, Fischer & Brox, 2015*), which consists of a symmetric encoder–decoder and skip connections. With such an elegant structural design, Unet has achieved great success in medical image processing. Along this technical line, many algorithms have been developed for various types of medical image segmentation. The excellent performance of these Unet-based methods in medical image segmentation has demonstrated the strong ability of CNN to learn features. However, the inherent localization and weight sharing of the receiver domain in convolutional operations make it difficult for CNN-based methods to learn explicit global information and remote semantic information interactions (*Xie et al., 2021*), which to some extent cannot meet the stringent requirements for segmentation accuracy in the field of medical image segmentation. Many researchers have noticed this problem and designed some modules to solve it, including residual learning (*He et al., 2016*), dense connections (*Huang et al., 2017*), self-attention mechanisms (*Schlemper et al., 2019*; *Wang et al., 2018*) and image pyramids (*Zhao et al., 2017*). Nevertheless, these methods still have some limitations and cannot explicitly model dependencies over long-distance and often exhibit sub-segmentation results. Applying Transformer to computer vision (*Wang et al., 2021b*; *Han et al., 2021*; *Zheng et al., 2021*) can alleviate the long-distance dependencies to some extent compared to other traditional CNN-based methods. At the same time, Transformer has a powerful global relationship modeling capability that has yielded amazing results in medical image analysis tasks. *Dosovitskiy et al. (2020)* proposed Vision Transformer (ViT) for image recognition tasks. Taking 2D image patches with location markers as input and pre-trained on large datasets, ViT achieves comparable performance to CNN-based methods. Many Transformer-based architectures have been proposed in the field of medical image segmentation, such as Trans-Unet (*Chen et al., 2021b*), Swin-Unet (*Cao et al., 2023*), ConViT (*d'Ascoli et al., 2021*) and ScaleFormer (*Huang et al., 2022*).

Many researchers have demonstrated the great potential of the structure on ViT-based image analysis (*Azad et al., 2023*; *Dalmaz, Yurt & Çukur, 2022*; *Li et al., 2021a*) and also promoted the study of Multi-Layer Perceptron (MLP) structures, such as MLP-Mixer (*Tolstikhin et al., 2021*), gMLP (*Liu et al., 2021a*), RepMLP (*Ding et al., 2022a*) and CycleMLP (*Chen et al., 2021c*). Despite the notable advancements in image analysis tasks, particularly in medical image segmentation, challenges persist when employing current methods that rely on transformer and MLP: (1) the network only accepts a fixed image size, and it is necessary to divide the image into a fixed size, which may not capture the fine-grained spatial details of the image; (2) it will inevitably cause boundary artifacts when applied to larger images (*Chen et al., 2021a*); (3) it lacks detailed positioning information because the input is considered as a one-dimensional sequence and only global information is modeled at all stages. When conducting semantic analysis, these models may lack the ability to accurately localize regions of interest (*Ni et al., 2022*). These problems are the

shortcomings of Transformer and MLP compared to CNN in extracting the underlying features and visual structure.

To solve the above-mentioned problems, we propose a Mixture Conv-MLP Network with Multi-scale Features Fusion Unet for Medical Image Segmentation (MCNMF-Unet). In each module, this network combines the advantages of convolution in extracting low-level features and visual structure and the advantages of MLP in fusing local and global information. The core of MCNMF-Unet is the Conv-MLP module, which combines the encoder and decoder characteristics in a U-shaped network and fuses MLP and convolution in each layer, enabling it to leverage the benefits of both methods. In this network, the key technology is the MLP Cross Gating (MCG) Block and the multi-axis and multi-windows MLP (MsM) block. MCG fuses the convolutional block information of different nodes through two paths. MsM uses multi-axis and multi-window to capture local and global information from multiple dimensions. The branches are evenly divided by channel, and the information is combined using different mechanisms along the respective axes. The computational burden of MsM is linearly related to the size of the input feature map. MCNMF-Unet has high performance in medical image segmentation with smaller parameters and FLOPs. The main contributions of this article are as follows:

(1) In order to combine the advantages of convolution and MLP to improve segmentation accuracy, we designed a general framework for medical image segmentation called MCNMF-Unet using the U-shaped encoder–decoder architecture.

(2) A multi-axis and multi-windows MLP (MsM) module is designed to capture feature map information from multiple layers and multiple dimensions, whose input does not require cropping of images, can receive images of arbitrary size, and always has a global receptive field.

(3) A Conv-MLP module is developed to perform feature cross-fusion on the two outputs of the convolution module, which is also a multi-path and multi-information interaction.

(4) MCNMF-Unet improves segmentation performance while maintaining a lightweight structure. Compared with most Unet-based improved networks, the performance of parameters and FLOPs indicators is better. It can achieve better segmentation results with less computing time and space spent.

# RELATED WORK

## CNN of U-shaped semantic segmentation methods

Most of the early semantic segmentation research was based on traditional machine learning algorithms of contours and regions (*Zhang et al., 2022*; *Tsai et al., 2003*). In recent years, with the development of deep learning, the Unet architecture, inspired by the fully convolutional neural network (FCN), has rapidly become the baseline network for computer vision tasks due to its elegant design and superior performance. Based on Unet's U-shaped architecture, various networks have been proposed for semantic segmentation. *Zhou et al. (2019)* proposed the Unet++ network,which integrates Unet structures of different scales into one network, captures features at different layers, and

integrates them into a shallower Unet structure by feature superposition, resulting in smaller scale differences in the feature maps during fusion. Based on the inspiration from deep residual learning (ResNet), *Zhang, Liu & Wang (2018)* proposed ResUnet. This architecture uses a series of stacked residual units as the basic building blocks, allowing for effective deepening of the network training layers. *Oktay et al. (2018)* proposed Attention Unet. The network proposes an attention gates (AGs) mechanism, which implicitly generates soft region suggestions and highlighting salient features useful for a specific task. By suppressing features of irrelevant regions, the sensitivity and accuracy of the model for dense label prediction are improved. *Çiçek et al. (2016)* proposed 3D Unet and introduced the structure into the field of 3D medical image segmentation.

## Vision Transformers

Transformers, first proposed by *Vaswani et al. (2017)*, is applied to natural language processing (NLP) tasks. Its architecture consists of stacked multi-headed self-attentive and feedforward MLP layers, which capture non-local interactions between words. In the field of NLP, the method has achieved optimal performance in various tasks. Inspired by the success of Transformers, *Dosovitskiy et al. (2020)* pioneered its introduction into computer vision and proposed ViT, which is the first pure Transformer architecture for image recognition. Recently, many researchers have tried to introduce ViT into medical image processing tasks and developed many Transformer-based models (*Heidari et al., 2023*; *Huang et al., 2021*; *Wang et al., 2021a*). *Chen et al. (2021b)* proposed TransUnet, which integrates the advantages of Transformers and Unet. TransUnet uses the local information encoded by CNN and the global context encoded by Transformer to encode tokenized image blocks from the CNNs feature map as an input sequence for extracting the global context. The decoder upsamples the encoded features and then combines them with the high-resolution CNN feature map for accurate localization. *Cao et al. (2023)* proposed Swin-Unet inspired by the shift window mechanism of Swin Transformer (*Liu et al., 2021b*), and their work is the first pure U-based architecture based on Transformer, where the encoder, bottleneck and decoder are built based on Swin-Transformer blocks. In the encoder, self-attentiveness from local to global is implemented; in the decoder, global features are upsampled to the input resolution for the corresponding pixel-level segmentation prediction.

## MLP vision

Recently, many researchers have been considering the necessity of a self-attention mechanism with a tremendous amount of computation in the Transformer architecture in computer vision. *Tolstikhin et al. (2021)* proposed the MLP-Mixer architecture (Mixer), an architecture developed entirely based on multilayer perceptrons (MLPs), which enables the interaction of two input dimensions through the interleaving of channel-mixing MLPs and token-mixing MLPs. MLP-Mixer architecture completely replaces the self-attention mechanism in ViT. *Liu et al. (2021a)* proposed gMLP, which is constructed from a basic MLP layer with gating. As an alternative to Transformer without self-attentiveness, gMLP consists only of channel and spatial projections with static parameterization. By comparing experimental results, it has been proven that the self-attention mechanism is

not important for ViT, and gMLP can achieve the same accuracy in critical language and vision applications. *Ding et al. (2022a)* proposed RepMLP, which incorporates local prior knowledge into a fully connected (FC) layer while reducing the number of parameters and inference time. The architecture takes full account of the global representation capability of the fully connected (FC) layer and the local capture property of the convolutional layer, using convolution to strengthen FC and make it possess locality and globality. With the rapid development of MLP, many researchers have applied it to medical image segmentation tasks. For instance, *Valanarasu & Patel (2022)* proposed UNeXt, a network based on convolution and MLP. The network uses convolutional blocks with a small number of filters in the initial and final blocks, and TokMLP blocks in the bottleneck section, effectively reducing the computational load while still modeling good representations. Similarly, *Jiang et al. (2023)* proposed MC-DC, which effectively fuses multilevel features of MLP and CNN using dual-path complementary modules. This network utilizes a global feature fusion module and a cross-scale local feature fusion module to reconstruct global and local information.

## METHOD

The proposed model is a novel architecture combining the advantages of convolution and MLP for medical image segmentation. In this section, we will introduce the backbone network of MCNMF-Unet and describe its main component modules. These modules are integral parts of the MCNMF-Unet network, and they include the Encoder Conv-MLP (ECM), Decoder Conv-MLP (DCM), Bottleneck Conv-MLP (BCM), Output Conv-MLP (OCM), MLP Cross Gating (MCG) and multi-axis and multi-windows MLP.

### The main backbone of MCNMF-Unet

The main framework of MCNMF-Unet follows the most classical U-like architecture, as shown in Fig. 1. For medical image segmentation, many studies have demonstrated that the Unet-like network is still a very competitive infrastructure (*Wang et al., 2022*; *Wu et al., 2023*; *Azad et al., 2022*). Unet is composed of various operations, including convolution, downsampling, upsampling and skip connections. The contracting path, located on the left side of the network, extracts semantic information from the image, reduces the image resolution, and enlarges the receptive field. On the other hand, the expanding path aims to predict pixel-wise segmentation, accurately locate target positions, and restore the image to a size similar to the input image. To capture lost detailed information from the contracting path, skip connections are added between the contracting and expanding paths. Most of the research on Unet-like networks is based on the CNN method, which has great advantages in extracting underlying features. However, its local receptive field and long-distance weak dependence still hinder the accuracy of complex structure segmentation. In addition to the advantages of Unet, an excellent segmentation network should also introduce MLP, which can increase the global receptive field of the network. Based on this idea, Encoder Conv-MLP (ECM) module, Decoder Conv-MLP (DCM) module, Bottleneck Conv-MLP (BCM) module and Output Conv-MLP (OCM) module are designed to build the U-shaped network. The MLP Cross Gating (MCG) in the ECM, DCM, BCM and

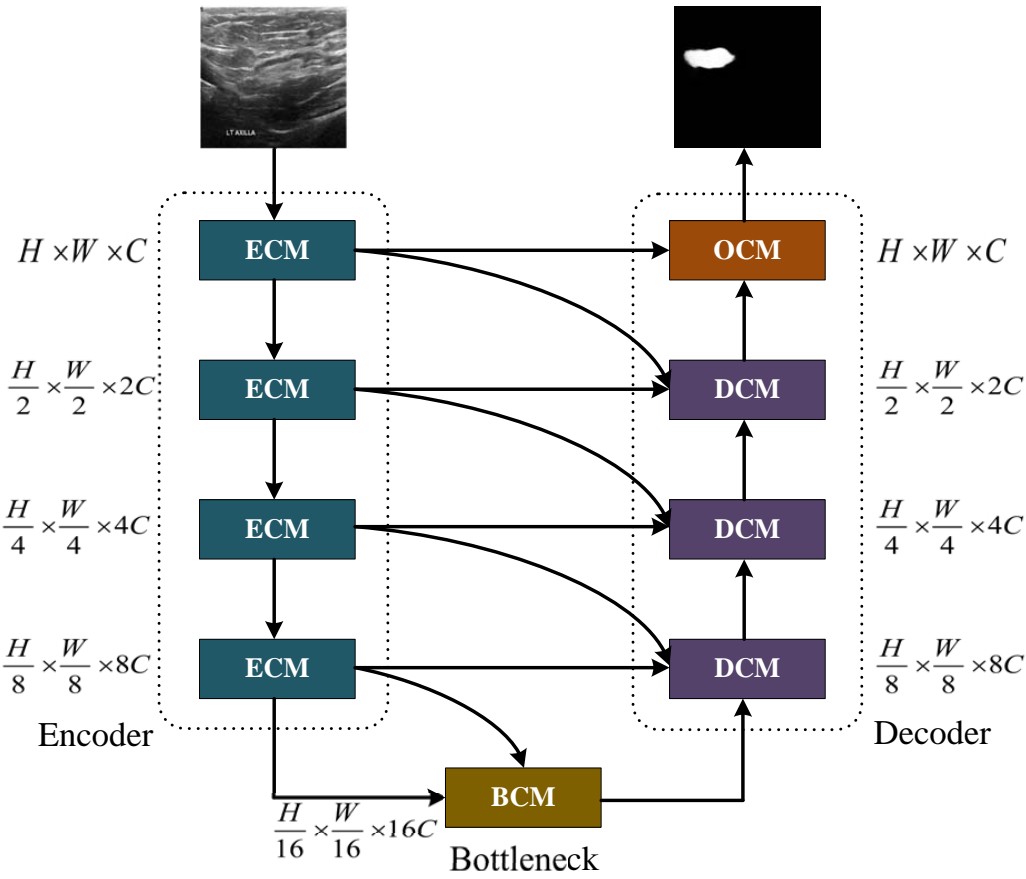

**Figure 1** **The MCNMF-Unet architecture.** Image source credit: BUSI, https://www.kaggle.com/datasets/aryashah2k/breast-ultrasound-images-dataset, CC0.

OCM is utilized to fuse information from two convolutional blocks. Two convolutional feature maps with local information are inputted into the MCG module, and through alternating Conv-MLP cross-fusion, they obtain multi-scale global receptive fields on local features. The convolutional blocks can alternately gate the information fusion of MLP. The MCG module offers a unique approach to information fusion, enabling the integration of global and local features through alternating Conv-MLP cross-fusion. This design of MCG differs from most information fusion methods, such as Unet++ and MultiResUnet, which solely rely on convolutional fusion and lack global sensory integration. TransUnet, on the other hand, only fuses transformers at the bottleneck layer, limiting the scope of global information fusion. In DC-MC, the information extraction of MLP is not influenced by convolution, and the global and local information extraction cannot mutually supervise each other, resulting in only a single convolutional feature being processed per fusion. The DCM, BCM and OCM performs skip connection with the ECM. Each DCM module will connect with two different ECM modules, and the OCM module and BCM module will connect with one ECM module. MCNMF-Unet architecture adopts an end-to-end training strategy that relies on a hybrid model design for each block, adaptively acquiring

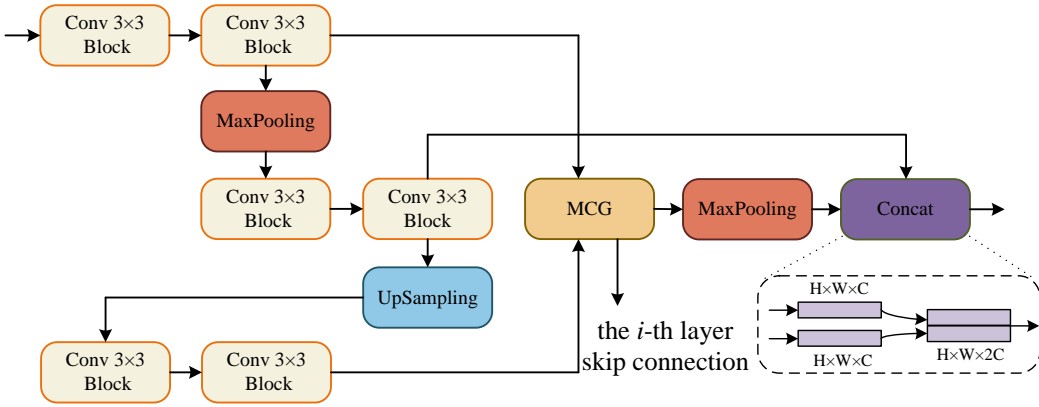

**Figure 2** Encoder Conv-MLP (ECM) module.

local and global sensory fields at each layer, where Conv is used locally and MLP is used for long-distance interaction, which can make full use of all feature information. For more detailed information, please refer to the section on the multi-axis and multi-windows MLP.

## The crucial sub-modules

The core sub-modules of MCNMF-Unet are ECM, DCM, BCM and OCM, which form the encoder and decoder of the U-shaped network. The Conv $3 \times 3$ block consists of Conv $3 \times 3$, BN and ReLU blocks. The ECM block references the encoder design concept of a standard U-shaped network, and considering the problem of information loss during encoder downsampling, the convolutional block downsampling and then upsampling operation is used, so that the information of the next layer can be obtained and at the same time as little information can be lost as possible, as shown in Fig. 2. In addition, the output of two convolutional blocks is obtained at the same layer and enters the MLP Cross Gating module (MCG), which cross-fuses the information from the convolutional blocks. The output of the MCG block serves as the input for skip connections with the decoder. Additionally, it is downsampled by a factor of two to reduce its size. Consequently, this feature map matches the size of the output feature map from the previously downsampled convolutional block. The two paths can then be concatenated in the channel dimension and utilized for subsequent module operations.

In the DCM, the goal is to maximize the restoration of image resolution. To achieve this, we incorporate upsampling and downsampling operations using convolutional blocks. This allows us to obtain more information at a larger resolution from the previous layer, making the current layer more forward-looking and capable of perceiving more detailed information. Additionally, in each layer of DCM, we have a convolutional block that is combined with the MCG block of the decoder. This fusion process helps to recover more information. Moreover, each DCM module is fused with two decoders from different layers, enabling the recovery of even more information. The decoder is specifically designed to feed two convolutional blocks from the same layer into the MCG module for further cross-fusion. This process captures global contextual information. The output of this module is

then upsampled by a factor of two to match the size of the upsampled convolutional block's output. Finally, the two paths are concatenated in the channel dimension and utilized for the next operation. In the BCM block, we follow the same design as DCM. The only difference is that the skip connection of the i-th layer is missing, but the skip connection of the i-1-th layer still retains, that is, the information has been gradually restored at the bottleneck layer, as shown in Fig. 3.

In the OCM module, the two convolutional blocks are connected serially and their output is fed to the MCG module for information fusion, followed by a $1 \times 1$ convolutional block yields the final segmentation result, as shown in Fig. 4.

## MLP cross gating

In order to effectively integrate the information from convolution and MLP, we used the MLP Cross Gating (MCG) module, as shown in Fig. 5. This module can well preserve the ability of convolution operation to extract local and fine features, and also extend the feature map to the global receptive field. MCG receives two inputs, which are the outputs of two different convolution blocks. Let the two inputs be $U_1$ and $V_1$, $U_1, V_1 \in \mathbb{R}^{H \times W \times C}$, then $U_2$ and $V_2$ can be obtained as:

$$U_2 = \sigma(Ds(LN(U_1))) \tag{1}$$

$$V_2 = \sigma(Ds(LN(V_1))) \tag{2}$$

where $LN$ is the Layer Normalization, $Ds$ is the Dense layers, and $\sigma$ is the GELU activation. Next, $U_2$ and $V_2$ enter the multi-axis and multi-windows MLP modules, respectively, and the data is divided into four channels for processing, as shown in Eq. (3).

$$L_1, L_2, G_1, G_2 = S(\sigma(Ds(LN(X)))) \tag{3}$$

where $S$ is the Split 4 heads operation for the different axes. The output end is concatenated according to Eq. (4).

$$M(X) = Do(Ds([Block_1(L_1), Block_2(L_2), Global_1(G_1), Global_2(G_2)])) \tag{4}$$

where $[\cdot, \cdot, \cdot, \cdot]$ is the concatenation. $M(X)$ denotes the output of the multi-axis and multi-windows MLP modules and which is used for the fusion of the two paths:

$$U_3 = U_2 \odot M(V_2) \tag{5}$$

$$V_3 = V_2 \odot M(U_2) \tag{6}$$

where $\odot$ represents corresponding multiplication of corresponding elements, and obtains the output of each path through the residual connection with the input:

$$U_4 = U_1 + (\rho(Ds(U_3))) \tag{7}$$
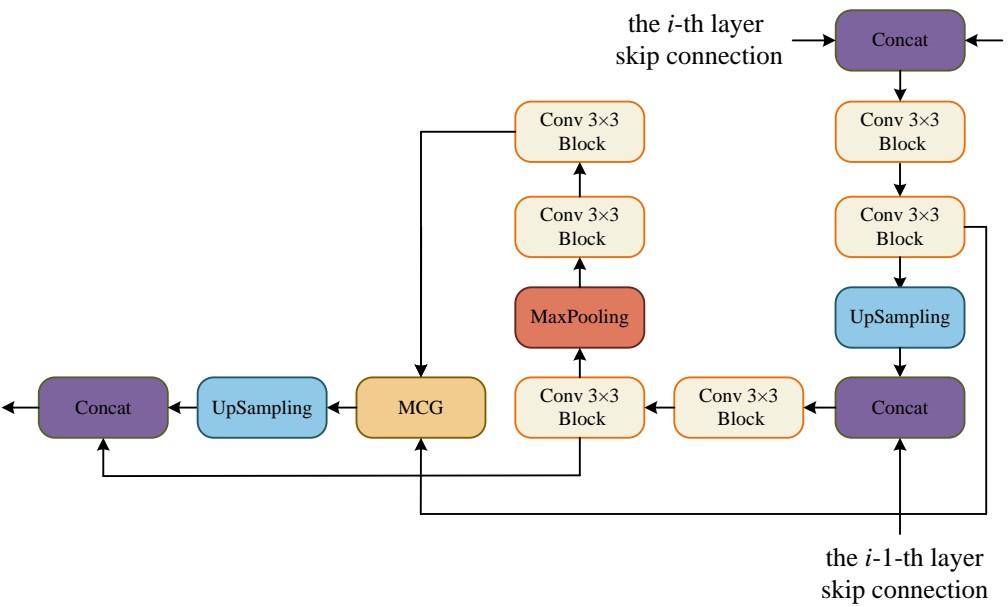

**Figure 3** Decoder Conv-MLP (DCM) module.

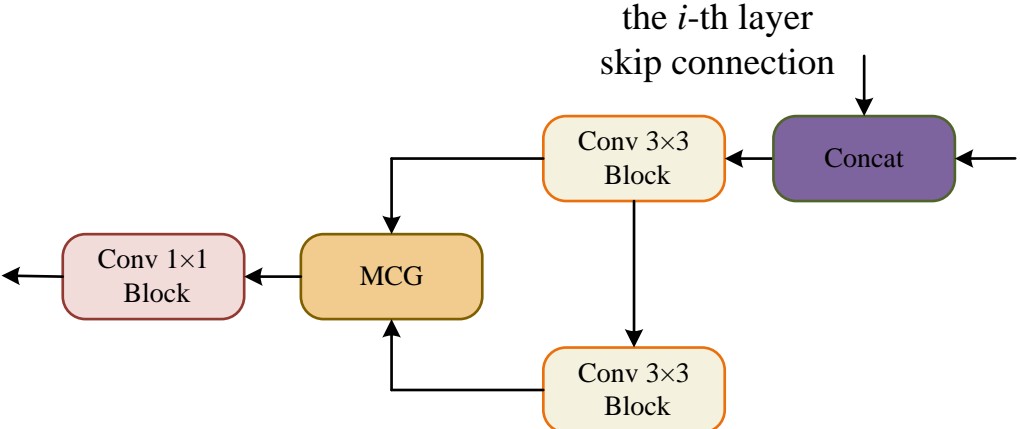

**Figure 4** Output Conv-MLP (OCM) module.

$$V_4 = V_1 + (\rho(Ds(V_3))) \tag{8}$$

where $\rho$ is the Dropout activation. To meet the system requirements and obtain an output of the system, we add the two paths together.

$$M = U_4 + V_4 \tag{9}$$

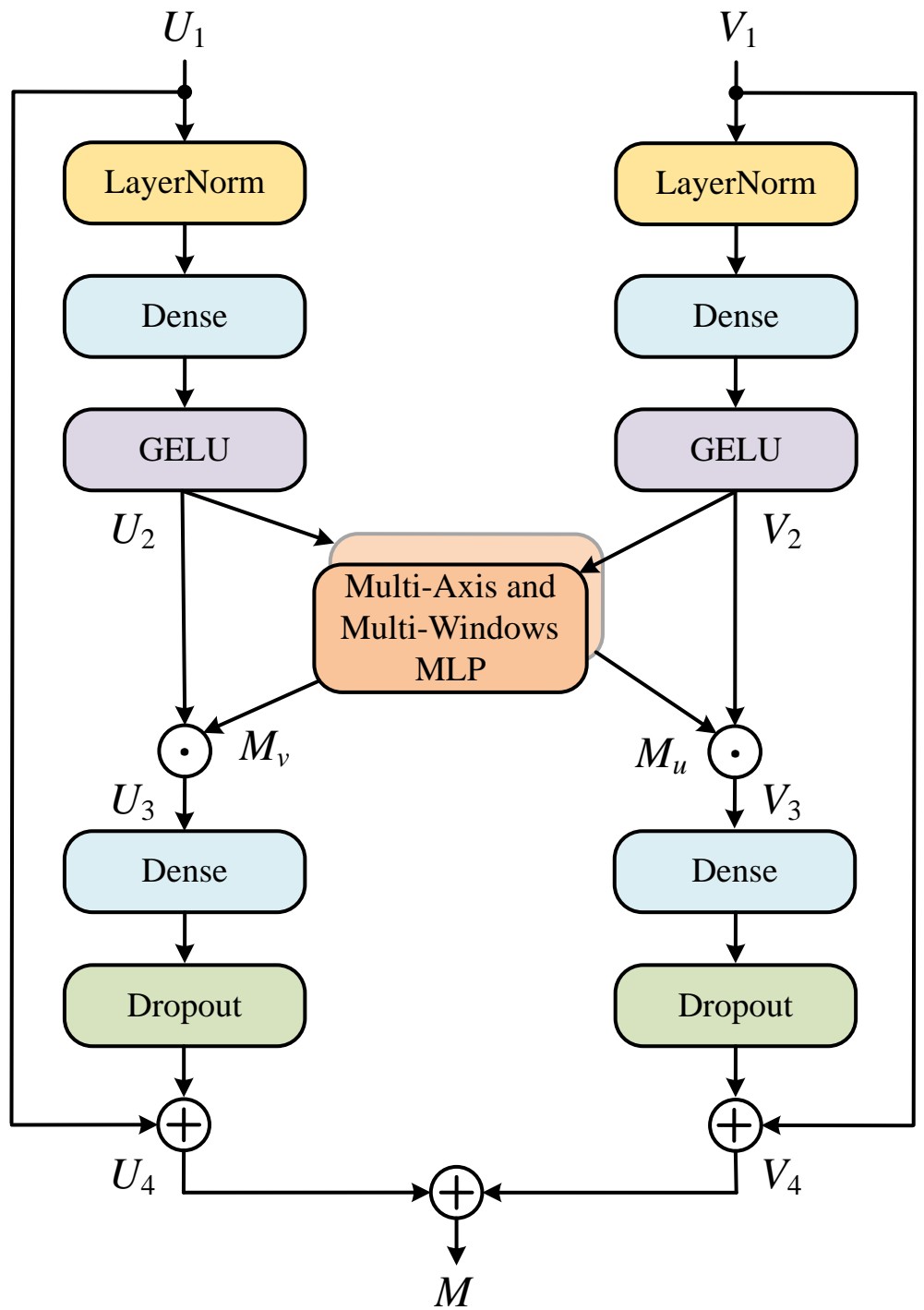

**Figure 5** MLP cross gating (MCG).

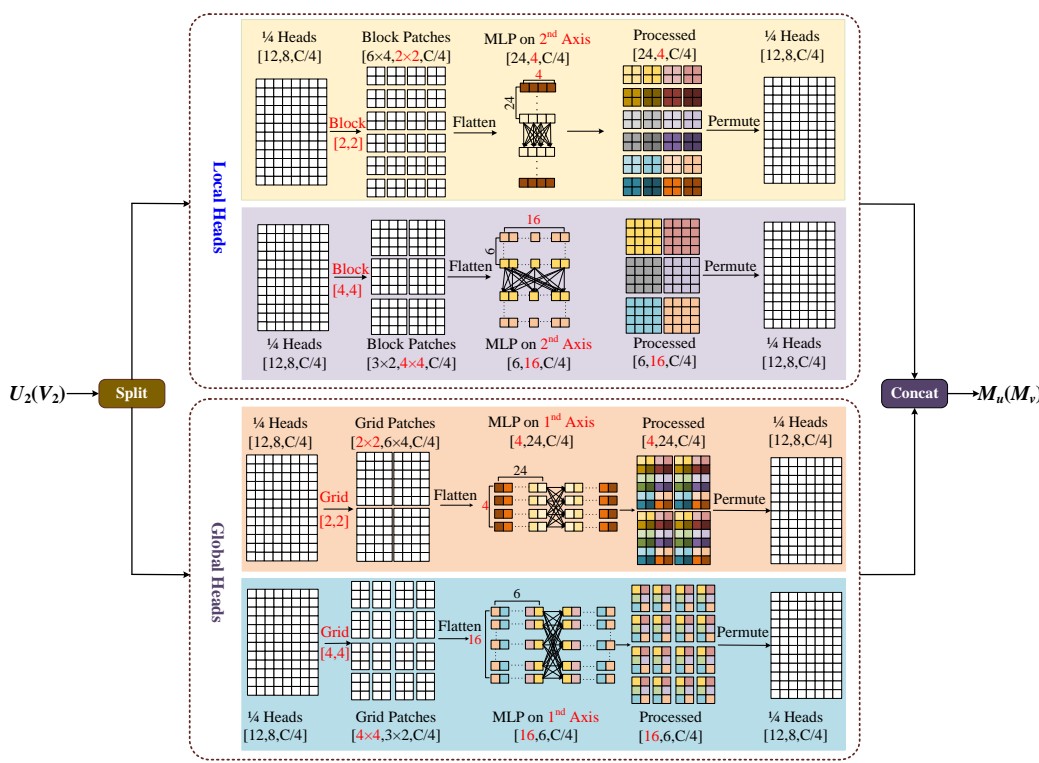

**Figure 6   Multi-axis and multi-windows MLP (MsM).**

## Multi-axis and multi-windows MLP

The multi-axis and multi-windows MLP (MsM) module is inspired by *Zhao et al. (2021)*, which proposes a sparse self-attention operation. The different forms of sparse self-attention on the two axes of the block image are performed, which can capture local and global information in parallel. However, the test images input by this module need a fixed size, which is easy to cause boundaries blur or artifacts, and is not friendly to model training for large-sized images. Based on *Tu et al. (2022)*, we proposed the multi-axis and multi-windows MLP framework, which established a basic framework for four-axis feature map information fusion, as shown in Fig. 6.

MsM module divides the input feature map into four heads by channel, and uses MLP in each head to fuse the corresponding information, two of which are sent to the local branch and the other two to the global branch. In the local branch, two heads are fed into two axes, each with an input tensor of $(H, W, \frac{C}{4})$. Meanwhile one of the axes is divided into $(H/2b \times W/2b)$ non-overlapping patches by window size $[b, b]$, and the tensor of $(\frac{H}{b} \times \frac{W}{b}, b \times b, \frac{C}{4})$ is obtained by blocking; the other axis is divided into $(H/2b \times W/2b)$ non-overlapping patches by window size $[2b, 2b]$, and the tensor of $(\frac{H}{2b} \times \frac{W}{2b}, 2b \times 2b, \frac{C}{4})$ is obtained by blocking. In the global branch, one of the axes uses a $[b, b]$ grid division to get window of $(\frac{H}{b}, \frac{W}{b})$ and the tensor of $(b \times b, \frac{H}{b} \times \frac{W}{b}, \frac{C}{4})$ obtained by griding; similarly the other axis uses a $[2b, 2b]$ grid division to get window of $(\frac{H}{2b}) \times (\frac{W}{2b})$ and the tensor of $(2b \times 2b, \frac{H}{2b} \times \frac{W}{2b}, \frac{C}{4})$ obtained by griding.

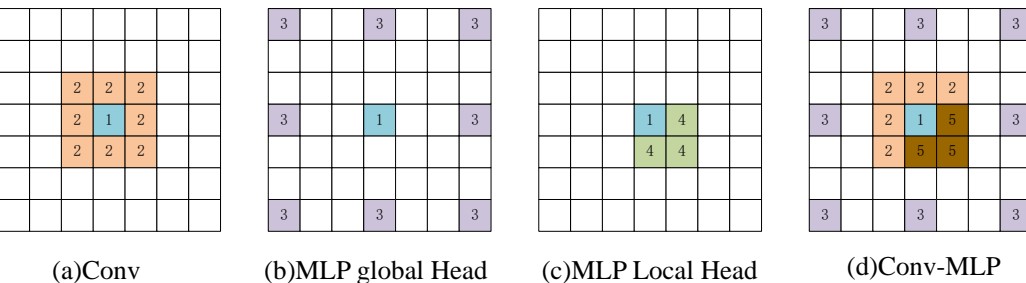

**Figure 7** (A–D) Visualization of the perceptual field for blue pixels.

In Fig. 6, we set $b = 2$ as an example, with the top half representing the local branch and the bottom half representing the global branch. As shown in Fig. 6, the input feature map is divided into 4-axis evenly by channel, where the one half goes into the local header and another half goes into the global header. In both local and global branches, different size block is used to divide and MLP is used for different levels of information mixing. Local branches correspond to local blending, global branches correspond to global blending, and the different colored squares in the diagram represent the degree of information blending within the windows. The final processed four-axis is concated and output. The module is designed to be completely end-to-end from input to output, without the need for specific size and cropping of the images, which will not have any negative effects due to cropping, and the complexity of the model is linear.

In Fig. 7, we used a $3 \times 3$ convolution and a window size of [2,2] to visually compare the MsM model and the effective receptive field of the blue pixel in the convolution (number 1). The convolution achieves local perception (orange, number 2). The MLP global branch achieves a globally sparse receptive field (purple, number 3), while the MLP local branch achieves a local receptive field (yellow, number 4). The brown pixels represent the common receptive field of the MLP global branch and the local branch (number 5). Our proposed Conv-MLP method not only achieves more detailed local interactions but also implements globally sparse interactions, effectively capturing local and global expansive spatial communication.

## EXPERIMENTS

### Datasets

The data collection process followed the methodology outlined in the study by *Yuan, Song & Fan (2023)*. The Breast Ultra Sound Images (BUSI) dataset (*Al-Dhabyani et al., 2020*) collected in 2018, contains breast ultrasound scans of 600 women aged 25 to 75. It includes 780 PNG format images cropped to remove unnecessary borders, with 647 images categorized as normal, benign, or malignant. Each image has its ground truth (masked image) and has been adjusted to $256 \times 256$ RGB size. The International Skin Imaging Collaboration (ISIC 2018) dataset (*Codella et al., 2019*; *Tschandl, Rosendahl & Kittler, 2018*) is the world's largest public dermoscope image library, containing 2594 images categorized as 20.0% melanoma, 72.0% nevus, and 8.0% seborrheic keratosis. The images

have various resolutions and have been adapted to $512 \times 512$ RGB size. The CVC-ClinicDB (*Bernal et al., 2015*) is a data set of colonoscopy images composed of endoscope images. These images are extracted from the video sequence of the colonoscopy. The Ground Truth image consists of a mask corresponding to the area covered by the polyp in the image. So the segmentation task only considers the images of polyps, a total of 612 images from 31 colonoscopy sequences were obtained. We resized all images to $256 \times 256$ RGB images.

## Implementation details

The experiments were carried out using a Python 3.8 framework and an RTX3090 (24GB) graphics card. A batch size of 8 was utilized, and the training process spanned 100 epochs. To enhance the diversity of training samples and bolster the model's generalization capability, the datasets were randomly combined multiple times in a 7:2:1 ratio for the training set, validation set, and test set. Data augmentation techniques, such as random image rotation, hue and brightness adjustments, and cropping, were also employed.

MCNMF-Unet employs a loss function that combines Binary Cross-Entropy and Dice coefficients. The loss function is described as:

$$BCELoss(Y,\hat{Y}) = -\sum_{i=1}^{N} Y(x_i) \cdot log\,\hat{Y}(x_i) \tag{10}$$

$$DiceLoss(Y,\hat{Y}) = 1 - \frac{2 \cdot |Y \cap \hat{Y}|}{|Y| + |\hat{Y}|} \tag{11}$$

$$L = 0.5 BCELoss(Y,\hat{Y}) + DiceLoss(Y,\hat{Y}). \tag{12}$$

In the context of image segmentation, where $Y$ and $\hat{Y}$ represent the true image label and the predicted image, and $x_i$ is the i-th pixel point of the image, $|Y|$ and $|\hat{Y}|$ denote the number of elements in the ground truth and predicted masks.

To assess the performance of our proposed framework compared to the baseline approach, we employ the F1_score coefficient and the Intersection over Union (IoU) coefficient as evaluation metrics. The F1_score is commonly used in classification problems and serves as a measure in binary or multi-classification scenarios. It is a harmonic average of the accuracy rate and recall rate, with a range of 0 to 1. The F1_score is calculated using the formula shown in Eq. (13).

$$F_1\_score = 2 \cdot \frac{Y \cdot \hat{Y}}{Y + \hat{Y}} \tag{13}$$

The IoU score is a standard metric for evaluating object segmentation tasks. It measures the similarity between the predicted and ground truth regions of the objects in a set of images. The IoU score is defined by the formula shown in Eq. (14).

$$IoU = \frac{|Y \cap \hat{Y}|}{|Y| \cup |\hat{Y}|} \tag{14}$$

**Table 1  Performance comparison of different networks in the BUSI, ISIC2018 and CVC-ClinicDB datasets.**

| Networks | Params (in MB) | GFLOPs | BUSI | | ISIC 2018 | | CVC-ClinicDB | |
|---|---|---|---|---|---|---|---|---|
| | | | IoU (%) | F1_score (%) | IoU (%) | F1_score (%) | IoU (%) | F1_score (%) |
| Unet (2015) | 31.13 | 55.84 | 63.54 | 77.14 | 78.42 | 87.46 | 79.12 | 88.17 |
| Res-Unet (2018) | 62.74 | 94.56 | 63.81 | 77.46 | 79.02 | 87.95 | 77.94 | 87.44 |
| Attention-Unet (2018) | 51.99 | 56.95 | 69.22 | 81.50 | 79.61 | 88.32 | 81.25 | 89.48 |
| Unet++ (2019) | 9.16 | 34.75 | 65.78 | 79.05 | 78.88 | 87.81 | 78.77 | 87.95 |
| MultiResUnet (2020) | 7.25 | 18.60 | 63.29 | 77.15 | 79.18 | 88.07 | 77.39 | 86.63 |
| Trans-Unet (2021) | 105.32 | 38.52 | 67.12 | 80.91 | 80.49 | 88.72 | 80.48 | 89.92 |
| UNeXt (2022) | **1.47** | **0.58** | 66.95 | 79.37 | 80.57 | 88.95 | 74.59 | 84.69 |
| MedFormer (2022) | 28.07 | 22.43 | 66.68 | 79.63 | 81.30 | 89.46 | 81.48 | 89.74 |
| MC-DC (2023) | 85.30 | 100.97 | 72.57 | 82.89 | 80.62 | 88.53 | 82.42 | 90.12 |
| MCNMF-Unet | 33.25 | 53.80 | **74.19** | **84.89** | **81.99** | **89.96** | **84.04** | **91.18** |

**Notes.**
Bold values indicate the optimal value of each evaluation index.

## Verification of model performance

In this section, the effectiveness of the MCNMF-Unet method is validated through comparisons with established medical image segmentation algorithms and popular models such as Unet (*Ronneberger, Fischer & Brox, 2015*), Res-Unet (*Zhang, Liu & Wang, 2018*), Attention-Unet (*Oktay et al., 2018*), Unet++ (*Zhou et al., 2019*), MultiResUnet (*Ibtehaz & Rahman, 2020*), Trans-Unet (*Chen et al., 2021b*), UNeXt (*Valanarasu & Patel, 2022*), MedFormer (*Gao et al., 2022*) and MC-DC (*Jiang et al., 2023*).

The evaluation of the MCNMF-Unet model is mainly compared in terms of the number of parameters, computational complexity and segmentation accuracy. Table 1 shows the comparison results. Compared with the base line model (Unet), params only increased by 2.12M, and GFLOPs decreased by 2.04. Notably, our proposed network has significantly fewer parameters compared to TransUnet, with a reduction of 72.07M. This reduction leads to a substantial decrease in memory usage during the training process. In terms of validating segmentation performance, we compare IoU and F1_scores in three public datasets: BUSI, ISIC 2018 and CVC-ClinicDB.

MCNMF-Unet shows the best performance on all three datasets. In the BUSI dataset, the IoU and F1_scores of MCNMF-Unet reached 74.19% and 84.89%, respectively. Compared with the results of the other six models, the IoU coefficients increased by 1.60%~10.90%. the F1_scores increased by 1.00%~7.46%. In the ISIC 2018 dataset, MCNMF-Unet achieved IoU and F1_scores of 81.99% and 89.96% respectively, and improved the IoU coefficients by 0.69%~3.57% compared to the other six models. the F1_score improved by 0.50%~2.50%. In the CVC-ClinicDB dataset, the IoU and F1_scores of 84.04% and 91.18%, respectively. The IoU coefficients improved by 1.62%~9.45%. The F1_score improved by 1.06%~6.49%. The comparative results indicate the better segmentation performance of MCNMF-Unet.

As the results of the evaluation metrics in Table 1, MCNMF-Unet achieves the best results, but visual observations are needed to determine whether the proposed model

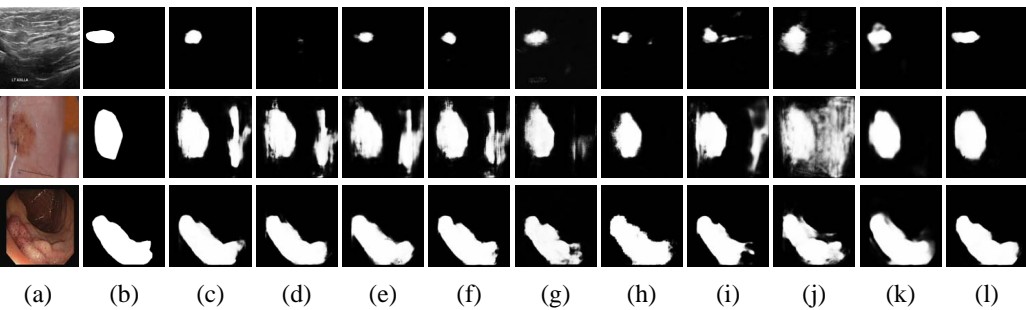

(a)   (b)   (c)   (d)   (e)   (f)   (g)   (h)   (i)   (j)   (k)   (l)

**Figure 8** **Qualitative comparisons.** Row 1: BUSI dataset, Row 2: ISIC 2018 dataset, Row 3: CVC-ClinicDB. (A) Input, (B) Ground Truth, (C) Unet, (D) Unet++, (E) Res-Unet, (F) Attention-Unet, (G) MultiResUnet, (H) Trans-Unet, (I) UNeXt, (J) MedFormer, (K) MC-DC and (L) MCNMF-Unet. Image source credits: Row 1, BUSI, https://www.kaggle.com/datasets/aryashah2k/breast-ultrasound-images-dataset, CC0; Row 2, ISIC 2018 Task 1 Lesion Segmentation (0016016), https://challenge.isic-archive.com/data/#2018, CC0; Row 3: CVC-ClinicDB, https://polyp.grand-challenge.org/CVCClinicDB, First column (C) Hospital Clinic, Barcelona, Spain, Second column, (C) Computer Vision Center, Barcelona, Spain.

works as expected. To this end, In Fig. 8, we also give examples of visual comparisons of the segmentation in BUSI, ISIC 2018 and CVC-ClinicDB.

MCNMF-Unet demonstrates superior performance compared to other widely used networks, as evidenced by its ability to effectively capture finer segmentation details. The results also indicate that MCNMF-Unet exhibits greater resilience in complex scenes, minimizing the occurrence of unexpected segmentation outcomes.

## Ablation experiments

We conduct extensive ablation experiments to verify the performance of MCNMF-Unet. All experimental evaluations are performed on the BUSI dataset. The experiment verifies the superiority of our proposed module while controlling other variables the same.

### Importance of MCG

In the experiment, the MCG module is replaced with convolution and attention modules (replacing the corresponding modules in Fig. 2, Fig. 3 and Fig. 4), and the details are shown, including the Params and GFLOPs after replacing these modules, as well as the IoU and F1_scores on the BUSI dataset. We mainly used the convolution module (Conv), Position Attention Module (PAM), Channel Attention Module (CAM), Convolutional Block Attention Module (CBAM) and Coordinate Attention Module (CoordAM). The data in Table 2 shows that the MCG module still achieves the best segmentation performance while keeping the learnable parameters of the model and the lowest computational complexity.

### Effects of Multi-axis and Multi-windows MLP approach

The experiments also further explored the impact of the proposed Multi-axis and Multi-windows MLP approach. As shown in Table 3, we mainly made changes to the Multi-axis and Multi-windows MLP module (Fig. 6) and conducted comparative experiments between the single-axis Block branch (Block), the single-axis Grid branch (Grid), the single-axis

**Table 2 The impact of MCG modules on network performance.**

| Variant | Params | GFLOPs | IoU (%) | F1_score (%) |
|---|---|---|---|---|
| Conv | 38.43 | 57.79 | 67.20 | 79.84 |
| CAM | 38.43 | 58.86 | 71.40 | 83.10 |
| PAM | 38.98 | 257.21 | 71.58 | 83.68 |
| CBAM | 34.62 | 52.94 | 70.71 | 82.42 |
| Coord.AM | 34.96 | 52.94 | 68.82 | 78.61 |
| **MCG (Ours)** | **33.25** | **51.80** | **74.19** | **84.89** |

**Notes.**
Values in bold indicate the experimental results in the proposed network in this article.

**Table 3 Effect of multi-axis and multi-windows MLP on network performance.**

| Variant | IoU (%) | F1_score (%) |
|---|---|---|
| Block | 72.67 | 83.89 |
| Grid | 72.80 | 83.94 |
| Block-Grid | 70.97 | 82.64 |
| Grid-Block | 69.00 | 81.36 |
| 2-Axis | 73.47 | 84.48 |
| **MsM (Ours)** | **74.19** | **84.89** |

**Notes.**
Values in bold indicate the experimental results in the MCNMF-Unet.

Block followed by Gride serial branch (Block-Grid), the single-axis Grid followed by Block branch (Grid-Block), the two-axis Block-Grid branch(2-Axis) and Multi-axis and Multi-windows MLP (MsM). Through these experiments, it can be seen that the Multi-axis and Multi-windows MLP approach achieves the optimal segmentation results, with the IoU and F1_score improvement reaching 0.72%–5.19% and 0.41%–3.53%, respectively.

# CONCLUSIONS

In this article, we design a novel deep neural network architecture based on U-shaped structure for medical image segmentation. The core idea is to integrate the advantages of CNN and MLP into the coding and decoding layers through MLP Cross Gating module, while Multi-axis and Multi-windows MLP module can capture the feature map information from multiple windows and multiple dimensions with global receptive field always. Our proposed MCNMF-Unet is essentially an improved network based on Unet, which can be effectively modeled with a small amount of parameters and computation. The experimental results show that MCNMF-Unet outperforms the state-of-the-art baseline on various benchmarks for the segmentation of biomedical images from different types. Moreover, our network achieves very excellent results for image segmentation with complex backgrounds. However, MCNMF-Unet architecture represents a preliminary exploration of the benefits of combining convolution and MLP. There is still room for further research to explore more scientific and effective ways of information fusion between these two components. Additionally, the current work primarily focuses on 2D medical image segmentation. However, with the prevalence of 3D medical images, it is crucial to extend

the proposed method to handle segmentation tasks in three-dimensional space. For future work, we plan to explore more scientific and effective ways of information fusion between convolution and MLP, and extend our method to 3D medical image segmentation.

### Funding

This work is supported by the Natural Science Foundation of Fujian Province (Grant Nos. 2020J01816, 2022J01916) and the Principal Foundation of Minnan Normal University (Grant No. KJ18010). There was no additional external funding received for this study. The funders had no role in study design, data collection and analysis, decision to publish, or preparation of the manuscript.

### Grant Disclosures

The following grant information was disclosed by the authors:
The Natural Science Foundation of Fujian Province: 2020J01816, 2022J01916.
The Principal Foundation of Minnan Normal University: KJ18010.

### Competing Interests

The authors declare there are no competing interests.

### Author Contributions

- Lei Yuan conceived and designed the experiments, performed the experiments, analyzed the data, performed the computation work, prepared figures and/or tables, authored or reviewed drafts of the article, and approved the final draft.
- Jianhua Song conceived and designed the experiments, analyzed the data, prepared figures and/or tables, authored or reviewed drafts of the article, and approved the final draft.
- Yazhuo Fan performed the experiments, performed the computation work, prepared figures and/or tables, and approved the final draft.

### Data Availability

The code is available at Zenodo: faithlei1. (2023). faithlei1/MCNMF-Unet: Second release of MCNMF-Unet (v1.0.1). Zenodo. https://doi.org/10.5281/zenodo.10050510.

The datasets are available at:

- Breast Ultra Sound Images (BUSI), https://www.kaggle.com/datasets/aryashah2k/breast-ultrasound-images-dataset (CC0: Public Domain).

- CVC-ClinicDB, https://www.kaggle.com/datasets/balraj98/cvcclinicdb, taken from https://polyp.grand-challenge.org/CVCClinicDB/.

- International Skin Imaging Collaboration (ISIC 2018) https://challenge.isic-archive.com/data/#2018

## Supplemental Information

Supplemental information for this article can be found online at http://dx.doi.org/10.7717/peerj-cs.1798#supplemental-information.

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
