# Peer review of "MCNMF-Unet: a mixture Conv-MLP network with multi-scale features fusion Unet for medical image segmentation"

_PeerJ Computer Science, doi:10.7717/peerj-cs.1798_

## Round 0.1 · original submission · Major Revisions

The reviewers have substantial concerns about this manuscript. The authors should provide point-to-point responses to address all the concerns and provide a revised manuscript with the revised parts being marked in different color.

Reviewer 1 ·

Basic reporting

This paper introduces MCNMF-Unet, a novel medical image segmentation model combining convolution and MLP for enhanced feature fusion. Tested on multiple datasets, it outperforms existing models. However, there are issues before publication.

Experimental design

Line 155-157: To enhance clarity and readability, consider providing a brief overview of the U-net-like network's principle and functionality before diving into its competitiveness. This will help readers unfamiliar with the U-net structure to have a foundational understanding before proceeding.

Line 164: The usage of MLP Cross Gating (MCG) to fuse information across convolutional blocks is introduced but not elaborated upon. It would be beneficial to provide more detailed insight into how MCG functions and its unique advantages compared to other fusion techniques.

Line 170: The distinction between the roles of Conv (for local applications) and MLP (for long-distance interactions) is mentioned. To enhance clarity, consider providing concrete examples or scenarios illustrating these roles, which will offer readers a more tangible grasp of their respective functionalities.

Validity of the findings

Please compare more peer experiments on Unet.

·

Basic reporting

This work offers a comprehensive overview of the evolution of U-Net-based models in medical image segmentation. It identifies limitations of transformer-based models for this purpose. The proposed solution, a combination of convolutional and MLP networks with a multi-scale feature fusion U-Net model, seamlessly integrates local and global information. Additionally, it effectively trims down the number of parameters and simplifies computational complexity.

Experimental design

1. Concat Module in Figures 2, 3, and 4:
- It would be immensely helpful if you could provide a detailed explanation of how the `concat` module operates in Figures 2, 3, and 4. While I assume it involves a straightforward addition of two vectors with the same dimension, a demonstration of the network structure would enhance clarity and understanding.

2. Layer Normalization in MCG Module:
- I'm curious about the rationale behind utilizing Layer Normalization (LN) in the MCG module. Given that LN is more commonly associated with NLP tasks to account for varying input lengths, it would be enlightening to understand the specific benefits in this context. Additionally, did you perform a comparison between LN and Batch Normalization (BN) in terms of performance?

3. Dataset Split Ratio:
- In the datasets section, it would be immensely beneficial to explicitly state the ratio for training, validation, and testing data. I check the code and observed a 0.7/0.3 split for training and validation sets, but without the test set. Introducing a test set, alongside loading parameters from the best epoch, could boost confidence in the model's capabilities and guard against overfitting concerns.

4. Cross-Validation Usage:
- Is cross-validation incorporated into the experimental setup. If not, is this due to limitations in computational resources?

Validity of the findings

1. Clarification on Loss Function Parameters:
- It would greatly enhance clarity if the authors explicitly stated each parameter (e.g., Xi, Y, N) in the loss function equation. This would provide readers with a clear understanding of the variables involved.

2. Comparison of Proposed Model with Recent Advances:
- While the introduction mentions some noteworthy models from recent years (such as ConViT in 2021, ScaleFormer in 2022, and Swin-Unet in 2023), the evaluation focuses solely on older models predating 2021. This approach potentially undermines the strength of the results and conclusions, as it does not demonstrate how MCNMF-Unet performs against the most current state-of-the-art models. Considering the author's assertion that MCNMF-Unet surpasses the state-of-the-art baseline, a comparative analysis against recent advances would significantly strengthen the credibility of the findings

Additional comments

Please ensure to include a README file in the GitHub repository. Additionally, providing clear information on package dependencies is crucial for ensuring the replicability of the experiments.

Reviewer 3 ·

Basic reporting

The study of "MCNMF-Unet: a mixture Conv-MLP network with multi-scale features fusion Unet for medical image segmentation" proposed a combined neural network frame for medical image segmentation by using CNN, multi-axis and multi-windows cross gate MLP, and U-Net. However, there are some items need to be revised and improved.

Experimental design

There’re also other similar works using CNN and MLP for image segmentation. You should include them in “RELATED WORK”, and compare your results with them in “EXPERIMENTS”.

Validity of the findings

Line 197, “then U2 and V1 can be obtained as”, here “V1” should be “V2”.
Line 261, “compared with the most popular TransUnet, Params is more than three times smaller with a reduction of 72.07M” should be rephrased.
In line 283, what’s “Section 4.2”?
Figure 7, Row 1 should be BUSI dataset, Row 2 should be ISIC dataset.
Table 3, “IOU” should be “IoU”.

Additional comments

In “ABSTRACT”, multi-axis and multi-windows MLP should be added, which is an important part of this manuscript.
In “CONCLUSIONS”, cross gate MLP and limitation of your research should be included.

---

## Round 0.2 · Minor Revisions

There are some remaining minor concerns that need to be addressed.

Reviewer 1 ·

Basic reporting

After careful review of the content, all previous comments have been addressed. The paper is now ready for publication.

Experimental design

After careful review of the content, all previous comments have been addressed. The paper is now ready for publication.

Validity of the findings

After careful review of the content, all previous comments have been addressed. The paper is now ready for publication.

·

Basic reporting

The author's response has addressed nearly all of the questions I raised, which is commendable. However, there are a few areas where minor revisions could further enhance the paper, making it even more compelling for publication standards. Implementing these changes would contribute to the overall strength and clarity of the work.

Experimental design

Regarding the Preliminary Experiments:
The author's mention of conducting preliminary experiments by substituting the LN layer with a BN layer in the MCG module is quite intriguing. To strengthen the paper's argument and provide more convincing evidence for the suitability of LN, I suggest including the results of these experiments in the supplementary material. Detailed presentation of these findings would offer valuable insights and further substantiate the choice of LN over BN.

Validity of the findings

In Table 1, since the author lists various models along with the year they were proposed in previous materials, I recommend a slight modification for improved clarity. It would be beneficial to include the respective years in parentheses immediately following each network name in the first column. This formatting change would enhance readability and provide a clearer chronological context for each model mentioned in the table

Reviewer 3 ·

Basic reporting

All my concerns have been addressed. However, in line 381-382, added limitation part "However ...... a preliminary exploration" needs to be revised because of syntax errors. Then, the manuscript is ready to be published after correcting this small syntax error.

Experimental design

The study has detailed and reasonable experimental design.

Validity of the findings

Results show better performance compared with related works based on metrics and images.

---

## Round 0.3 · accepted · Accept

All reviewers' comments have been addressed, and I recommend accepting this manuscript.